# The Difference of Sea Level Variability by Steric Height and Altimetry in the North Pacific

**Qianran Zhang [1], Fangjie Yu [1,2,\*] and Ge Chen [1,2]**

[1] College of Information Science and Engineering, Ocean University of China, Qingdao 266100, China; zqr@stu.ouc.edu.cn (Q.Z.); gechen@ouc.edu.cn (G.C.)

[2] Laboratory for Regional Oceanography and Numerical Modeling, Qingdao National Laboratory for Marine Science and Technology, Qingdao 266200, China

\* Correspondence: yufangjie@ouc.edu.cn; Tel.: +86-0532-66782155

**Abstract:** Sea level variability, which is less than ~100 km in scale, is important in upper-ocean circulation dynamics and is difficult to observe by existing altimetry observations; thus, interferometric altimetry, which effectively provides high-resolution observations over two swaths, was developed. However, validating the sea level variability in two dimensions is a difficult task. In theory, using the steric method to validate height variability in different pixels is feasible and has already been proven by modelled and altimetry gridded data. In this paper, we use Argo data around a typical mesoscale eddy and altimetry along-track data in the North Pacific to analyze the relationship between steric data and along-track data (SD-AD) at two points, which indicates the feasibility of the steric method. We also analyzed the result of SD-AD by the relationship of the distance of the Argo and the satellite in Point 1 ($P_1$) and Point 2 ($P_2$), the relationship of two Argo positions, the relationship of the distance between Argo positions and the eddy center and the relationship of the wind. The results showed that the relationship of the SD-AD can reach a correlation coefficient of ~0.98, the root mean square deviation (RMSD) was ~1.8 cm, the bias was ~0.6 cm. This proved that it is feasible to validate interferometric altimetry data using the steric method under these conditions.

**Keywords:** steric height; sea level variability; interferometric altimeter validation

## 1. Introduction

Since the first radar altimetric satellites were launched in the 1970s, altimetric data have been widely used for understanding the ocean, including eddies and sea level change [1]. Since 1992, more than two radar altimetry datasets were able to be used at the same time. However, even when combining the data, it is difficult to observe ocean dynamic variability less than an ~150 km scale because of the traditional altimetry only has the along-track data [2]. Compared with traditional radar altimetry, new-generation interferometric altimetry, which is onboard satellites such as the Surface Water and Ocean Topography (SWOT) or "Guanlan" mission [3,4], can yield high spatial resolutions over two swaths to obtain sea level variability in two dimensions with unprecedented spatial resolution in the swath and give finer observations of the ocean phenomenon [5]. Meanwhile, to improve data accuracy, the calibration and validation (Cal/Val) of altimetry data are important. Fixed platforms, such as Harvest, Corsica, Gavdos and the Bass Strait, are widely used to validate radar altimetry along-track data [6–9]; this approach extrapolated the onshore sea surface height (SSH) out to the offshore nadir point with an accuracy of (1.88 ± 0.20) cm and a standard deviation of 3.3 cm, which suggested that the approach presented was feasible in absolute altimeter Cal/Val [10]. However, the validation of interferometric altimetry is not only for along-track data; it is a validation of spatial resolution such as validate sea level variability between the different pixels at the same time in a swath. Now, the research

on the Cal/Val methods of interferometric altimetry mainly focuses on the offshore method [11] and the steric method [12,13]. There is no perfect solution. The offshore method was developed to extend the single-point approach to a wider regional scale. In this paper, the study is based on the steric method, which uses the observed data to calculate the steric height and validate the sea level variability. Wang et al. [13] proved that using the steric method to validate the interferometric altimetry data by 20 glider and mooring data in the swath area was feasible based on the model data. However, the model data and the theoretical analysis are not sufficient to prove that it is feasible to validate sea level variability between different pixels in a swath by the steric method.

The variability in sea surface height observed by an altimeter has two causes: the steric signal and the non-steric height (NSH) signal [14]. The sea surface height (SSH) variability is largely dominated by the steric signal within a small area [14–16]. Meanwhile, the previous studies have greatly contributed to the relationship between the steric height (SH) and the sea level. Roemmich et al. [17] used the 10-yr SH increase and the 22-yr SSH increase data for analyzing the interannual trends in the South Pacific and found an increase difference of approximately 3 cm. Meyssignac et al. [18] studied the relationship between sea level and SH in different areas and proved that trends were largely dominated by SH except in high-latitude areas (> 60°N and < 55°S) and some shallow shelf seas. For seasonal and inter-annual period variability, Dhomps et al. [19] showed that sea level variability is clearly dominated by baroclinic motion. Regarding wind and rainfall, Perigaud et al. [20] used the model to analyze the influence of rainfall anomalies on sea level variability due to the SH. Song et al. [21] analyzed the wind force influence on the sea level variability and proved that the annual SSH variability is mostly due to steric changes. Therefore, the SH variability is an important part of sea level variability [22].

However, studying long-term scale and large areas of the sea level variability were not enough to explain the feasibility of using the steric method to validate the sea level variability in two dimensions. In this paper, we used field observations to obtain the relationship between steric data and along-track data (SD-AD) at two points. There were 17 Argo stations data for captured the mesoscale eddy and the altimetric along-track data to analyze the feasibility of using the SH to validate the interferometric altimetry data. We used the experimental results to analyze the SD-AD under different factors. The altimetry data, Argo data and the methods are introduced in Section 2. The results are presented in Section 3. The discussion is presented in Section 4 and is followed by conclusions.

## 2. Materials and Methods

### 2.1. Argo Profile

Argo, as the broad-scale global array of profiling floats, has already grown to be a major component of the ocean observing system. The Argo data can be used for measuring high-vertical-resolution temperature-salinity-pressure (T-S-P) profiles that, in turn, can be used to compute ocean density and SH for ocean monitoring and study [23]. The Argo was deployed by the rate of about 800 per year since 2000. Now, the global array has about 3000 floats and distributed approximately every 3 degrees (300 km). Floats can be observed at the depth up to 2000 m every 10 days, with 4–5-year lifetimes for individual instruments. In this paper, the data were downloaded from the French Research Institute for Exploitation of the Sea (IFREMER) Global Data Assembly Centres (GDAC) Server (ftp.ifremer.fr/ifremer/argo/), which has passed through real-time quality control procedures, and most of the data have passed through delayed-mode procedures.

In order to find the Argo data with a larger number in a small area to match the satellite data, we used data from 17 Argo stations (numbers 2901550 to 2901566), which capture the eddy south of the Kuroshio extension located east of Japan, as shown in Figure 1a. The eddy was detected by the Okubo–Weiss (OW) method using the AVISO gridded data [24], and the eddy information is also obtained from the Okubo-Weiss method like the eddy radius. The eddy passing area was getting from the eddy radius and the eddy center position. In this paper, the Argo was in operation for more than one year, starting in March 2014 and the longest running time reaching June 2015. The profile sampling

interval was one day, an anticyclonic eddy (AE) moved through the Argo track comprising more than 3000 hydrographic profiles following the AE [25]. The T-S-P profiles measured pressure at depths up to 1000 m, and the profiles were divided into more than 350 levels.

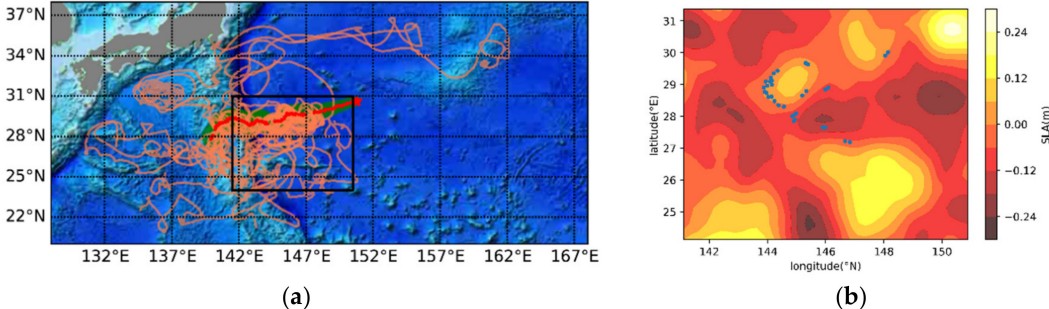

**Figure 1.** Argo and eddy data. (**a**) The coral lines are the Argo data with numbers from 2901550 to 2901566, the red line is the track of the eddy center, and the green area is the eddy passing area. The base map shows the depth of the ocean, and the sea level anomaly (SLA) of the black box is shown in Figure 1 (**b**). (**b**) The Argo position on May 15, 2014 and the base map show the performance of SLA.

### 2.2. Altimetry Data

In this paper, the altimetry along-track data were provided by Copernicus Marine Environment Monitoring Service (CMEMS) project, which includes the SLA and absolute dynamic topography (ADT) [26]. As shown in Figure 2, the different databases have the different reference level. The SSH is the sea level relative to the reference ellipsoid, which was obtained by the satellite altitude minus the altimetric range observed by the altimeter; the SLA was the sea level relative to the mean sea surface height (MSSH), and the ADT was the sea surface height above the Geoid [27]. Moreover, the reference ellipsoid is a regular ellipsoid that fits the earth with the equatorial radius and flattening coefficient. The geoid is a gravity equipotential surface that would correspond with the ocean surface if ocean was at rest, and the MSSH is the long-term sea surface height average. The altimetry data that we used were near real-time reanalysis along-track data with delivery times up to 6 months. Figure 3 shows the along-track data of which the spans of the two data were 14 km. With an objective to match the time of the Argo data, we selected the Jason-2 data from 2014 and 2015, which was derived from a satellite mission from 2008 to 2017.

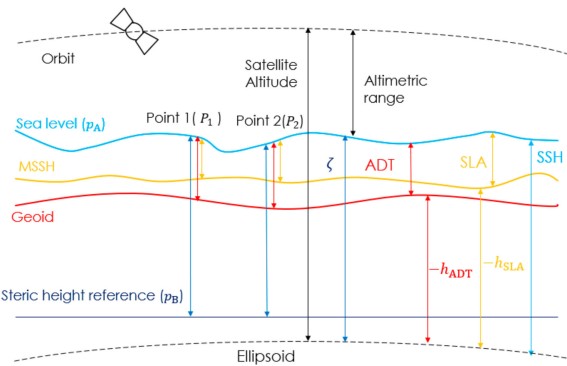

**Figure 2.** The relationships of sea level variability under different reference levels. The sea surface height (SSH) measured by the altimeter was the sea level reference for the reference ellipsoid, the sea level anomaly (SLA) was the sea level reference for the mean sea surface height (MSSH) and the absolute dynamic topography (ADT) was the sea surface height above the Geoid. The steric height (SH) was caused by the change of the density between the sea level and the reference level.

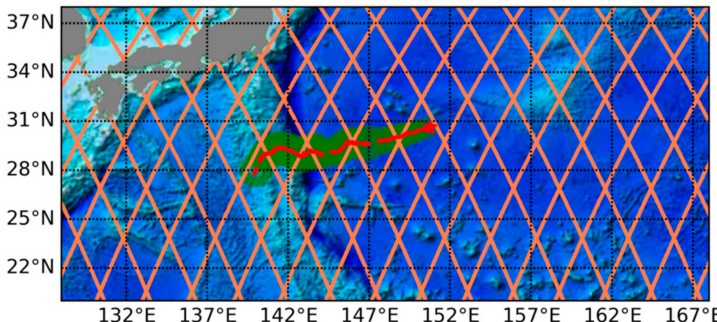

**Figure 3.** Satellite along-track data and eddy track data. The coral points represent the satellite position. The red points are the eddy center, and the red star is the start position of the eddy. The green areas are the eddy passing area.

### 2.3. Wind Data

In this paper, the wind data were provided by the European Centre for Medium Weather Forecasting (ECMWF) ERA-Interim data. The data assimilation system used to produce ERA-Interim is based on a 2006 release of the IFS (Cy31r2). The system includes a four-dimensional variational analysis (4D-Var) with a 12-hour analysis window. The spatial resolution of the data set is approximately 80 km (T255 spectral) on 60 vertical levels from the surface up to 0.1 hPa. In order to match the Argo data, we select the ERA-Interim data from 2014 and 2015, and the data were gridded data which size of 0.125 degrees, with one datum every 6 hours.

### 2.4. Method

In this paper, our aim was to determine the feasibility of using the SH to validate sea level variability between different pixels in the swath of the interferometric altimeter, and find the relationship between the height variability of the altimetry data and the SH variability at two points. To ensure the influencing factors of the relationship between the SD-AD, we conducted the theoretical analysis first.

As far as physical oceans are concerned, the hydrostatic equation is as follows:

$$\frac{\partial p}{\partial z} = -\rho g; \ \rho = \rho_0 + \rho\prime \tag{1}$$

where $p$ is the pressure, $\rho$ is the potential density, $\rho_0$ is the reference potential density (1027.5 kgm$^{-3}$), $\rho\prime$ is the potential density anomaly and $g$ is the gravity of Earth. The equation also can be written as follows:

$$p_{\mathrm{A}} - p_{\mathrm{B}} = -\rho_0 g(\zeta + h) - \int_{-h}^{\zeta} \rho\prime g dz \tag{2}$$

where $p_{\mathrm{B}}$ is the ocean bottom pressure, $p_{\mathrm{A}}$ is the atmospheric surface pressure, $\zeta$ is the SSH referenced to z = 0 and $-h$ is the depth of the ocean. After rearrangement, one obtains:

$$\zeta = \frac{p_{\mathrm{B}}'}{\rho_0 g} - \frac{p_{\mathrm{A}}}{\rho_0 g} - \int_{-h}^{0} \frac{\rho\prime}{\rho_0} dz \tag{3}$$

where $p_{\mathrm{B}}' = p_{\mathrm{B}} - \rho_0 g(\zeta + h)$ represents the bottom pressure anomaly and the term $\int_{0}^{\zeta} \frac{\rho\prime}{\rho_0} dz$ is neglected because $\zeta \ll h$. Therefore, the variable of $\zeta$ at two points is:

$$D_\zeta = \zeta_1 - \zeta_2 = \frac{p_{\mathrm{B}1}' - p_{\mathrm{B}2}'}{\rho_0 g} - \frac{p_{\mathrm{A}1} - p_{\mathrm{A}2}}{\rho_0 g} - \int_{-h}^{0} \frac{\rho\prime_1 - \rho\prime_2}{\rho_0} dz \tag{4}$$

In this paper, we used the SLA and the ADT to represent the $\zeta$ as the satellite data and the $p_2$ was the MSSH and geoid, respectively.

SH is not exactly the height (in meters) of an isobaric surface above the geopotential surface, which is Φ. The Φ is:

$$\Phi = \int_0^z gdz \qquad (5)$$

To calculate geostrophic currents, oceanographers use a modified form of the hydrostatic equation. The vertical pressure gradient (1) is written as follows:

$$\frac{\delta p}{\rho} = \alpha \delta p = -g\delta z; \alpha \delta p = \delta \Phi \qquad (6)$$

where Φ is the geopotential surface, $\Phi = \int_0^z gdz$, α is the specific volume and $SH = \Phi/g$ (in SI units) has almost the same numerical value as height in metres.

At Point 1 ($P_1$), the SH is:

$$SH = \frac{1}{g} \int_{p_B}^{p_A} \alpha(35,0,p)dp + \frac{1}{g} \int_{p_B}^{p_A} \alpha' dp \qquad (7)$$

where α (35, 0, p) is the specific volume of sea water with a salinity of 35, temperature of 0 °C and pressure *p*. In this paper, in order to ensure the consistency of the depth in different profiles, 900 meters is selected as the $p_2$ value in the SH equation. The second term $\alpha'$ is the specific volume anomaly. Therefore, the variable of *SH* at two points is:

$$D_{SH} = SH_1 - SH_2 = \frac{1}{g} \int_{p_B}^{p_A} \alpha'_1 - \alpha'_2 dp = -\int_{-h}^{\zeta} \frac{\rho'_1 - \rho'_2}{\rho_0} dz \qquad (8)$$

Thus, the difference between the SH and the satellite data is caused by the bottom pressure variable and the atmospheric surface pressure variable.

In order to improve the reliability of the results, we chose the SH and steric height anomaly (SHA) as the Argo data source to compare the satellite data (ADT and SLA) to study the feasibility of using the steric method to validate the interferometric altimetry. The mean steric height (MSH) was calculated by the World Ocean Atlas 2013 version 2 monthly fields (WOA 13). Thus, the difference between the two points is as follows:

$$D_{SLA} = SLA_1 - SLA_2 = SSH_1 - MSSH_1 - (SSH_2 - MSSH_2) = SSH_1 - SSH_2 - MSSH_1 + MSSH_2 \qquad (9)$$

$$D_{ADT} = ADT_1 - ADT_2 = SSH_1 - Geoid_1 - (SSH_2 - Geoid_2) = SSH_1 - SSH_2 - Geoid_1 + Geoid_2 \qquad (10)$$

$$D_{SH} = SH_1 - SH_2 = SSH_1 - NSH_1 - (SSH_2 - NSH_2) = SSH_1 - SSH_2 - NSH_1 + NSH_2 \qquad (11)$$

$$D_{SHA} = SHA_1 - SHA_2 = SH_1 - MSH_1 - (SH_2 - MSH_2) = SSH_1 - SSH_2 - NSH_1 + NSH_2 - MSH_1 + MSH_2 \qquad (12)$$

The difference between the two databases is mainly caused by the barotropic and the difference of the different reference planes. In this paper, the two databases were the satellite data and the Argo data. The satellite data include the SLA and the ADT, which got from the satellite along-track data, and the Argo data include the SH and the SHA. In order to eliminate the impact, we conducted the following study. We initially utilized the satellite along-track data position to match the Argo position based on distance and time; then, we obtained two points as a set of data for which the space-time interval between the Argo position and the satellite position are within a certain range. The set of data seemed to represent $P_1$ in equations 9–12, and the time matching was done again for each set of data to find point 2 ($P_2$). Then, we obtained a group of data that included the two satellite positions and two Argo position. According to the time information of this group of data, matching the eddy which was detected by the OW method like the eddy radius. As shown in Figure 4, each group of data had five points, including two Argo data points (blue cross-shaped symbol) as $P_1$ and $P_2$, respectively, two

AVISO data points (cyan dots) as $P_1$ and $P_2$, respectively, and an eddy centre point (red dot). Thus, we can got four height variables between $P_1$ and $P_2$: $D_{SH}$ and $D_{SHA}$, which were from the Argo points, and $D_{SLA}$ and $D_{ADT}$, which were from the altimetry data.

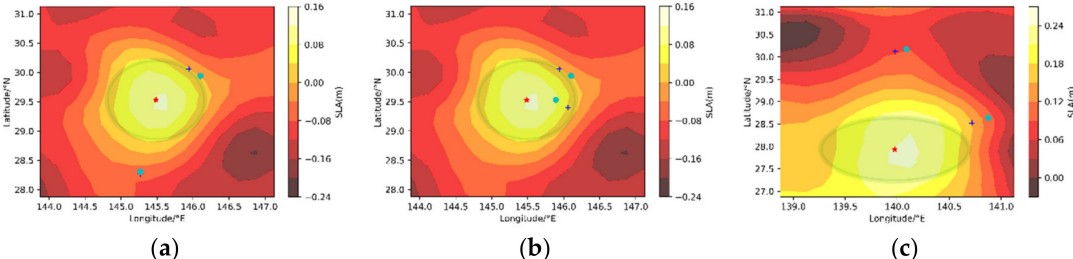

<div style="text-align:center">(<b>a</b>)         (<b>b</b>)         (<b>c</b>)</div>

**Figure 4.** The grouped data. Each map has a group of data in which the blue cross-shaped points are the Argo positions, the cyan dot points are along-track positions, the red star is the eddy center and the green circle areas are the eddy areas. The maps (**a**) indicate data groups No. 6, the maps (**b**) indicate data groups No. 13 and the maps (**c**) indicate data groups No. 24. They will be discussed in the next section.

In the next section, we discuss the distance between the Argo position and the along-track position and the influence of barotropy to analyzing the impact of different factors on the SD-AD data.

## 3. Results

### 3.1. Distances between the Argo Positions and the Along-Track Positions

With the distance between an Argo position and the satellite position being small, the Argo position and the satellite position can be approximately considered as lying on the same point, and the representative error is reduced. The result is shown in Figure 5 when we used the SH as the Argo database. With the selected data based on the distance between the Argo position and the satellite position less than 0.2 degrees in the latitude and longitude direction and the time being less than four hours, the trends are show in Figure 5a. The maximum difference of the SD-AD reach near 20 cm, and when the satellite database selects different, the SLA performs better than the ADT. To output a better result using the SH to validate the altimetry data, we used more restrictive conditions. As shown in Figure 5b, 26 groups of data were used for the analysis with the distance between the Argo position and the satellite position less than 0.15 degrees in the latitude and longitude direction and the time being less than four hours.

We analyzed the result of the SD-AD by using three different evaluation data; the results are shown in Table 1. When we see the result of the SD-AD under the control condition is 0.2 degrees and four hours, the results indicate that the results of the SD-AD by SLA are better than the result by ADT. When the control condition up to 0.15 degrees, the results of the SD-AD have greatly improved, especially in the bias. The data analysis under the control condition is 0.15 degrees, indicating that for the root mean square deviation (RMSD) and the correlation coefficient, the SLA was more suitable than the ADT, but were different when the bias was used as an evaluation condition. Furthermore, the difference between the SD-AD by ADT and SLA are more similar from the control condition of 0.2 degrees to 0.15 degrees. For the interferometric altimeter, the satellite data can be divided into 5 km * 5 km grids; thus, the observation equipment used for validation can be integrated into the data less than 5 km away from the satellite observation. In addition, the results of using the steric method to validate the sea level variability between different pixels in the interferometric altimetry swath are better than this test because in reality, the limited conditions are stricter than those in this experiment. However, to avoid the samples being too sparse, the distance between the Argo position and the satellite position less than 0.15 degrees in the latitude and longitude direction was chosen as the screening condition in this experiment.

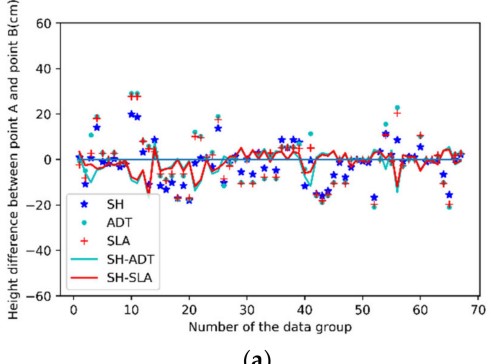
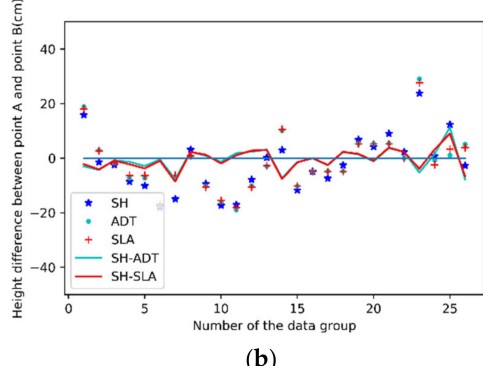

(**a**)                                                          (**b**)

**Figure 5.** The results of the height difference between the $P_1$ and $P_2$ by using the SH as the Argo data. The blue star is the height difference between SH at $P_1$ and $P_2$, the cyan dots are the height difference between ADT at $P_1$ and $P_2$, and the red crosses are the height difference between SLA at $P_1$ and $P_2$. The cyan line was the height difference between SD-AD (SH-ADT) at two points, and the red line was the height difference between SD-AD (SH-SLA) at two points. (**a**) The height difference between $P_1$ and $P_2$ where the distance between the Argo position and the satellite position less than 0.2 degrees in the latitude and longitude direction with a time difference less than four hours; (**b**) the height difference between the $P_1$ and $P_2$ where the distance between the Argo position and the satellite position less than 0.15 degrees in the latitude and longitude direction with a time difference less than four hours.

**Table 1.** The result of the SD-AD by the Argo data selecting the SH.

| Control Condition | Root Mean Square Deviation (RMSD) | | Bias | | Correlation Coefficient | |
|---|---|---|---|---|---|---|
| 0.2 degrees | ADT | SLA | ADT | SLA | ADT | SLA |
| 4 hours | 5.25 cm | 4.42 cm | 1.54 cm | 1.18 cm | 0.8858 | 0.9034 |
| 0.15 degrees | ADT | SLA | ADT | SLA | ADT | SLA |
| 4 hours | 4.07 cm | 3.87 cm | 0.54 cm | 0.57 cm | 0.9241 | 0.9292 |

When we used the SHA as the Argo database, we conducted the same experiment; the results are shown in Figure 6 and the evaluation data are show in Table 2. Compared with the control condition, it was 0.2 degrees in the latitude and longitude direction; the RMSD, bias and correlation coefficients are greatly improved when the control condition is 0.15 degrees in the latitude and longitude direction. Compared with the SD-AD of the Argo data selecting SH, the result of the SD-AD by the SHA decreased significantly. Moreover, the trend of SD-AD was similar between SD-AD results with two different Argo databases (SH and SHA), although the increase by the SHA is much larger than using the SH as the Argo database. The main reason is that the results of SD-AD by SHA are worse under the control condition of 0.2 degrees. Meanwhile, with the limit condition of the distance between the altimetry and Argo positions decreasing, regardless of the difference between the SHA and SH or the difference between the ADT and SLA, the SD-AD had a significant reduction.

**Table 2.** The result of the SD-AD by the Argo data select the SHA.

| Control Condition | RMSD | | BIAS | | Correlation Coefficient | |
|---|---|---|---|---|---|---|
| 0.2 degrees | ADT | SLA | ADT | SLA | ADT | SLA |
| 4 hours | 7.79 cm | 6.56 cm | 2.46 cm | 2.09 cm | 0.7393 | 0.7994 |
| 0.15 degrees | ADT | SLA | ADT | SLA | ADT | SLA |
| 4 hours | 5.57 cm | 5.42 cm | 0.54 cm | 0.57 cm | 0.8862 | 0.8941 |

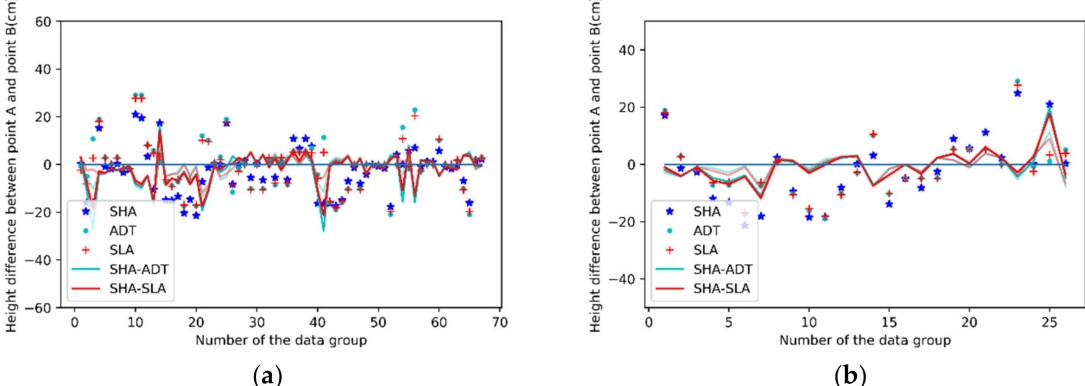

**Figure 6.** The results of the height difference between the $P_1$ and $P_2$ by using the SHA as the Argo data. The blue stars are the height difference between SH at $P_1$ and $P_2$, the cyan dots are the height difference between ADT at $P_1$ and $P_2$, and the red crosses are the height difference between SLA at $P_1$ and $P_2$. The cyan line was the height difference between SD-AD (SHA-ADT) at two point, and the red line was the height difference between SD-AD (SHA-SLA) at two point. The faded lines repeating lines from Figure 5 (for SH). (**a**) The height difference between $P_1$ and $P_2$ where the distance between the Argo position and the satellite position less than 0.2 degrees in the latitude and longitude direction with a time difference less than four hours; (**b**) the height difference between the $P_1$ and $P_2$ where the distance between the Argo position and the satellite position less than 0.15 degrees in the latitude and longitude direction with a time difference less than four hours.

To obtain better SD-AD results, we used more factors to analyze the results. At $P_1$, there is a distance between the Argo and the satellite. At $P_2$ there is still a distance between the Argo and the satellite. Figure 7 shows the relationships between the SD-AD as a function of the distance difference between these two distances. Regardless of the SH or SHA, the relationship of the SD-AD had a similar trend; moreover, no matter for the SH and SHA, the ADT data performed better than the SLA data when the distance difference was smaller. However, the results after linear regression were different in terms of the slope, the SHA slope larger than the SH one, thus, the SHA datasets had a clearer trend. In Figure 7b, with the distance different between the distance of the Argo and the satellite in $P_1$ and $P_2$ being larger than 13 km, the results became larger than 3 cm for the SHA case. However, there are some special set of data that the SD-AD performed poorly even when the distance difference between $P_1$ and $P_2$ was not very large, such as for No. 13. As shown in Figure 4b, the distance difference between $P_1$ and $P_2$ were similar; at this time, the directions of the Argo positions relative to the altimetry positions became another contributing factor. At $P_1$ and $P_2$, the angle difference between Argo position and satellite positions is close to 180 degrees. However, the sea level variability in the small range always occurred in the same direction. At $P_1$ and $P_2$, compared with the position of the Argo point, in which the satellite position in the different direction, the sea level variability was different. Therefore, making the distance difference between $P_1$ and $P_2$ similar and the direction between the Argo points and altimetry data points similar are an important condition when using the SD to validate the altimetry data.



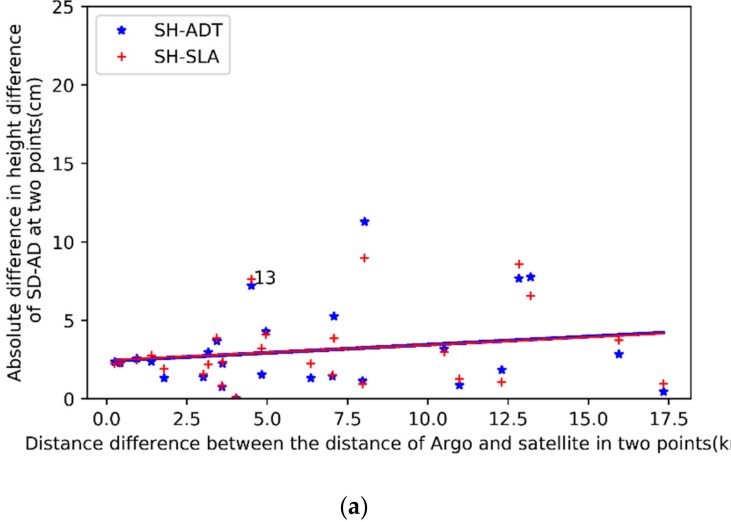

(**a**)

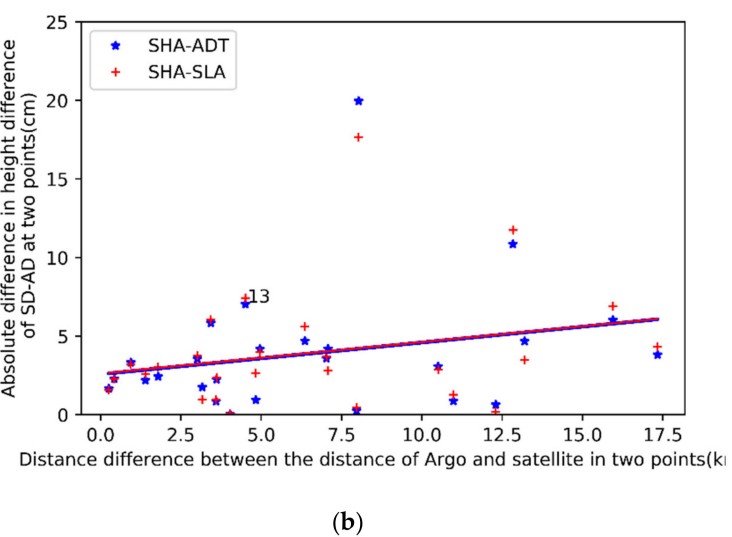

(**b**)

**Figure 7.** The relationship of the SD-AD as a function of the distance difference between the distance of the Argo and the satellite in $P_1$ and $P_2$, where (**a**) shows the relationship between the SH and the altimetry data; the blue stars show the absolute difference in the height difference between SD-AD (SH-ADT) at two points, and the red crosses show the absolute difference in the height difference between SD-AD (SH-SLA) at two points. (**b**) Shows the relationship between the SHA and the altimetry data; the blue stars show the absolute difference in the height difference between SD-AD (SHA-ADT) at two points, and the red crosses show the absolute difference in the height difference between SD-AD (SHA-SLA) at two points.

### 3.2. Barotropic Influence

Variability in the SSH can be decomposed into two contributions: barotropic and baroclinic [28]. Steric data can respond well to baroclinic changes, so in order to better reveal the effects of SD-AD, it is necessary to analyze the barotropic influence. In this paper, we analyzed the SD-AD by using three different approaches.

### 3.2.1. Distance between Two Points

The results of the SD-AD as a function of the distance between two Argo points are shown in Figure 8a,c. We can see that the distribution of the height difference in two points between the positive

and negative values is not consistent. And as the distance between the two Argo points became shorter, the results of the SD-AD under the satellite data selecting the ADT data or the SLA data are more similar. This proved that the geoid and MSSH were similar in a small range. With the increase in distance between the two Argo points, it is clear that the SD-AD increased and the result of the SD-AD has a significant linear relationship either selecting the ADT data or the SLA data as the satellite data (Figure 8b,d). When a linear regression analysis was performed on the distance between two Argo points, regarding the result of the SD-AD for the SH or SHA, the ADT data performed well when the distance between two Argo closer, despite the result after linear regression are basically consistent.

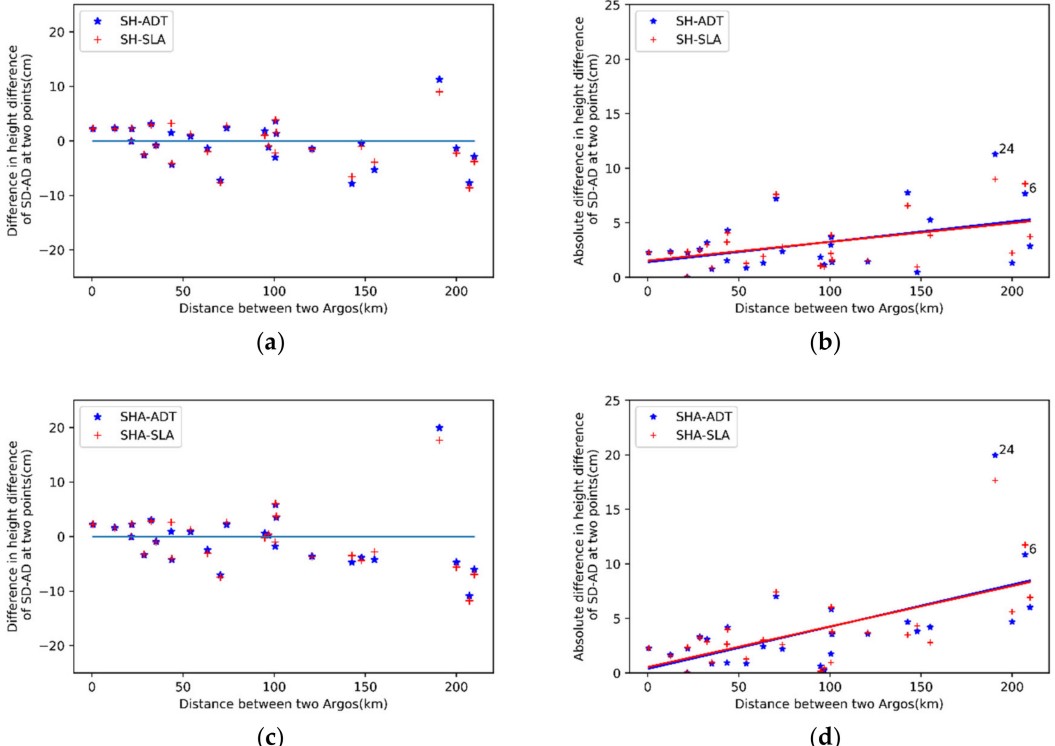

**Figure 8.** The relationship of the SD-AD as a function of distance between the two Argo points; the y axes of (**a**) and (**c**) are the difference at two points (A and B) of Argo data minus satellite data; the y axes of (**b**) and (**d**) represent the SD-AD. (**a**) and (**b**) are the result of the SD-AD by using the Argo data to select the SH; the blue stars show the difference in the height difference between SD-AD (SH-ADT) at two points, and the red crosses show the difference in the height difference between SD-AD (SH-SLA) at two points. (**c**) and (**d**) are the result of the SD-AD by using the Argo data to select the SHA; the blue stars show the difference in the height difference between SD-AD (SHA-ADT) at two points, and the red crosses show the difference in the height difference between SD-AD (SHA-SLA) at two point.

As shown in Figure 8, when the distance was large enough, the SD-AD data indicated that the values were not good enough to use the SH data to validate the altimetry data, as shown in group data No. 24 and group data No. 6. As shown in Figure 4, the distances between the two Argo points and the corresponding along-track points were 190 km and 207 km, which were too long to assume that the barotropic motion was similar. The trend of the SHA was clearer, which we can see in Figure 8d; with a distance larger than 120 km, it is hard to find a good result of the SD-AD by using the Argo data to select the SHA, which means that the non-steric influence significantly impacts the relationship between the steric height and the sea level.

### 3.2.2. Distances between the Argo Points and the Eddy Centre

There is a strong correlation between the mesoscale eddy and baroclinic instability [29]. Therefore, we used the sum of the distance of the Argo point and the eddy center in $P_1$ and $P_2$ to analyze the influence of the SD-AD. As shown in Figure 9, with the increase of the sum of the distance between an Argo point and the eddy center at two points, the SD-AD also shows an upward trend, and the selected ADT database as the altimetry data was better than the SLA database at smaller distances. The slope of the linear regression analysis result is similar between the SH database and SHA database but different in terms of the ADT and SLA.

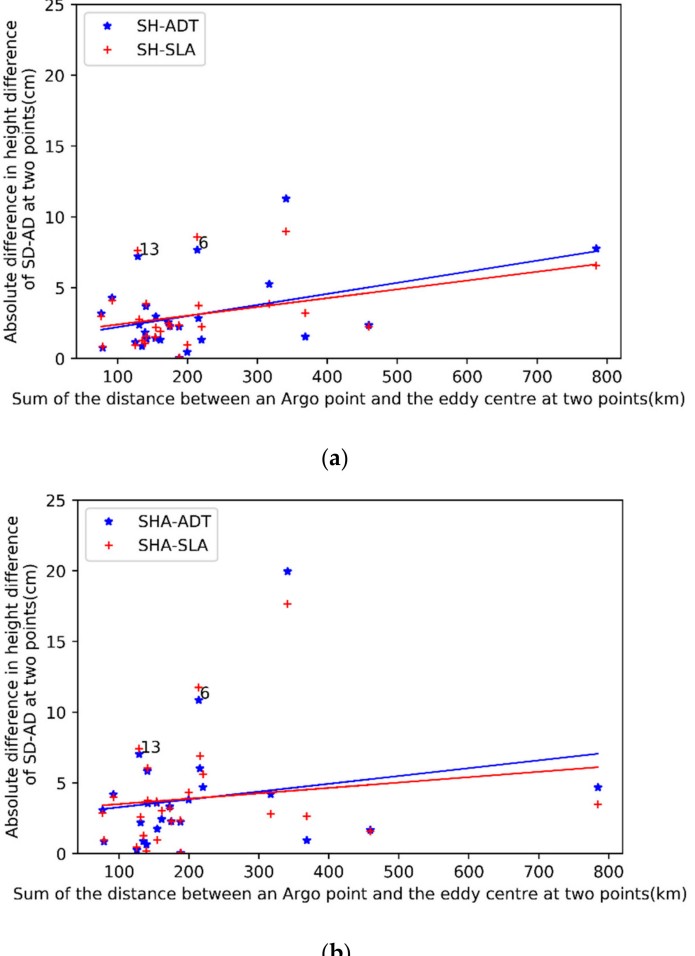

(**a**)

(**b**)

**Figure 9.** Shows the relationships of the SD-AD as a function of the sum of the distance between an Argo point and the eddy center at two points. (**a**) Shows the relationship between the SH and the altimetry data; the blue stars show the absolute difference in the height difference between SD-AD (SH-ADT) at two points, and the red crosses show the absolute difference in the height difference between SD-AD (SH-SLA) at two points. (**b**) The relationship between the SHA and the altimetry data; the blue stars show the absolute difference in the height difference between SD-AD (SHA-ADT) at two points, and the red crosses show the absolute difference in the height difference between SD-AD (SHA-SLA) at two points.

In Figure 9, with the sum of the distance between an Argo point and the eddy center at two points being larger, the difference between the ADT and the SLA was larger, thus, the MSSH and the geoid had greater differences under this condition. When we analyze the SD-AD data with distances smaller than ~200 km, the results perform well, except for group data No. 13 and group data No. 6; the reason why was introduced in the previous section. Moreover, it was also proven that the SD-AD is the result

of the multiple factors. Therefore, when the distance between the Argo points and the eddy center is smaller than 220 km, the results of the SD-AD are relatively concentrated, simply because the mesoscale eddy is a typical baroclinic phenomenon, the steric data can better represent sea level variations.

### 3.2.3. Wind Speeds

Barotropic ocean response to wind stress forcing is important for understanding sea level variability [30]. In this paper, considering that wind-forced motion is a type of barotropic motion and the wind force are related to the wind speed [31,32], we tried to find the relationship of the SD-AD by the wind speed. The range of the wind speed was from 1 m/s to 9.6 m/s for all points, and the maximum value of the wind speed difference was 1.3 m/s. As we can see from Figure 10, using the SHA provides clearer results than the SD-AD using SH. After the linear regression analysis accounting for the wind factor was performed, the SLA data initially performed better than the ADT data regardless of SH or SHA, although the slope of the SHA was greater than that of the SH. When the wind speed difference between two points was smaller than 1 m/s, the SD-AD results performed well in most cases.

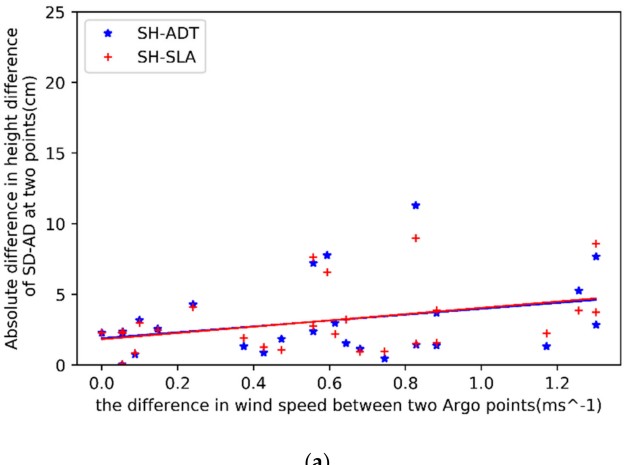

(**a**)

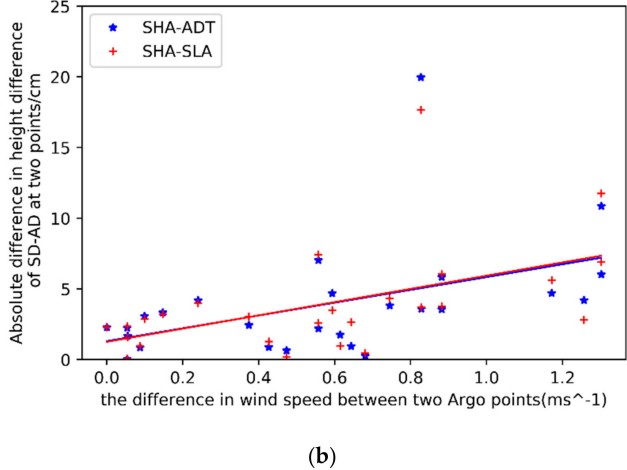

(**b**)

**Figure 10.** Shows the relationships of the SD-AD as a function of the difference in wind speed between two Argo points. (**a**) The relationship between the SH and the altimetry data; the blue stars show the absolute difference in the height difference between SD-AD (SH-ADT) at two points, and the red crosses show the absolute difference in the height difference between SD-AD (SH-SLA) at two point. (**b**) The relationship between the SHA and the altimetry data, the blue stars show the absolute difference in the height difference between SD-AD (SHA-ADT) at two points, and the red crosses show the absolute difference in the height difference between SD-AD (SHA-SLA) at two points.

### 3.3. Results under the Conditions

Considering the previous analysis in this paper, we used four conditions to limit the results to prove the influence of these factors, which included the distance difference between the distance of the Argo and the satellite in $P_1$ and $P_2$ being less than ~13 km and an angle difference of less than 120 degrees, the distance between two Argo points was less than ~120 km, the sum of the distance of the Argo point and the eddy center in $P_1$ and $P_2$ were less than ~220 km, and the difference in wind speed between two Argo points was less than ~1 m/s. There were seven groups of data after data control was applied which shown in Figure 11. The results are showed in Table 3. These values represent a further improvement with respect to previous analyses, and the results showed that sea level variability dominates by the baroclinicity in the small range. The results of these analyses proved that it is feasible to use the steric method to validate altimetry data.

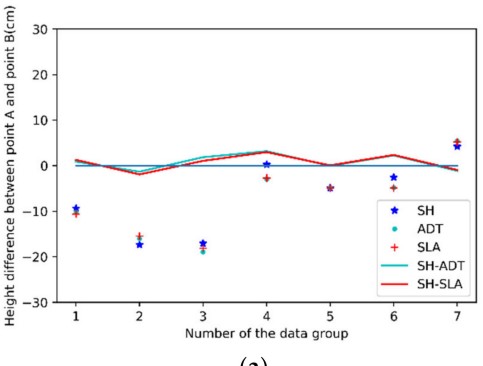
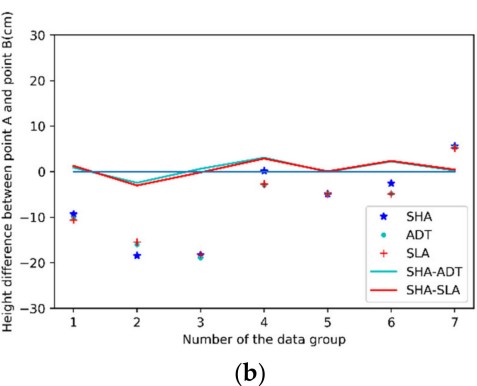

|  (a)  |  (b)  |

**Figure 11.** The SD-AD results after screening with the influence factor; (**a**) is the SH data selected as the steric data, where the blue stars are the height difference between SH at $P_1$ and $P_2$, the cyan dots are the height difference between ADT at $P_1$ and $P_2$ and the red crosses are the height difference between SLA at $P_1$ and $P_2$. The cyan line was the difference in the height difference between SD-AD (SH-ADT) at two points, and the red line was the difference in the height difference between SD-AD (SH-SLA) at two points. (**b**) is the SHA data selected as the steric data, where the blue stars are the height difference between SHA at $P_1$ and $P_2$, the cyan dots are the height difference between ADT at $P_1$ and $P_2$ and the red crosses are the height difference between SLA at $P_1$ and $P_2$. The cyan line shows the difference in the height difference between SD-AD (SHA-ADT) at two points, and the red line shows the difference in the height difference between SD-AD (SHA-SLA) at two points.

**Table 3.** The result of the SD-AD by four control conditions.

| Argo Database | RMSD | | BIAS | | Correlation Coefficient | |
|---|---|---|---|---|---|---|
| | ADT | SLA | ADT | SLA | ADT | SLA |
| SH | 1.79 cm | 1.76 cm | 0.81 cm | 0.69 cm | 0.9785 | 0.9780 |
| | ADT | SLA | ADT | SLA | ADT | SLA |
| SHA | 1.76 cm | 1.89 cm | 0.66 cm | 0.55 cm | 0.9841 | 0.9828 |

## 4. Discussion

We analyzed the relationships of the SD-AD from multiple perspectives. The first is the distance between the Argo position and the altimetry position. As the distance between the Argo position and altimetry data position decreases, the Argo data can better validate the result of the satellite data. However, when we validate the sea level variability between the different pixels in the swath of the interferometric altimeter, the influence of the distance between the Argo point and the altimetry position can be neglected, because the pixel from the interferometric altimeter is sufficiently smaller. Meanwhile, considering that the ocean variability has the same trend in a small area, the direction between the Argo and altimeter measurements sites becomes another important factor as shown

in Figure 4b. Thus, the effect of direction cannot be ignored when validating the interferometric altimetry data.

Barotropic influence is another important factor. Most of the ocean barotropic motions, such as barotropic tides and pressure- and wind-forced barotropic motions, drove mass variability over all frequency band studied here and yielded particularly strong variables at periods of 3–4 days, primarily at mid-high latitudes [33]. However, there are still some barotropic motions that are temporary and can cause the SH data to vary substantially. We used three different factors to analyze the SD-AD, deepening our understanding of the relationship of the SD-AD, which proved that the SD-AD is influenced by multiple factors, which include the distance difference between the distance of the Argo and the satellite in $P_1$ and $P_2$ and angle difference, the distance between two Argo points, the sum of the distance of the Argo point and the eddy center in $P_1$ and $P_2$ and the difference in wind speed between two Argo points.

Regardless of the changing factors, the slope of the satellite data selecting the ADT or SLA is similar under each factor except for the eddy center factor. Moreover, when the limit conditions were 0.2 degrees, the result of the SLA database was obviously superior to that of the ADT database. With the limit conditions becoming 0.15 degrees, the result of the ADT database performance improved. As shown in Figures 7–10, except for the influence of wind, the ADT data were better than the SLA data when the limit was more restrictive.

When comparing the SHA and the SH, when the limit conditions were 0.2 degrees, the result of the SH database was obviously superior to that of the SHA database. As shown in Figures 7–10, except for the influence of the eddy center, the slope of the SHA was larger than that of the SH which showed that the variability was more obvious and the SHA was influenced more by those factors. When the experimental conditions from the distance between the Argo position and the satellite position less than 0.2 degrees in the latitude and longitude direction and the time being less than four hours to the last 7 group data, the result of the RMSD, bias and the correlation coefficient are obviously better when the SHA is selected as the Argo data.

## 5. Conclusions

In this study, we evaluated the feasibility of using a steric method to validate the sea level variability between different pixels in a swath from multiple perspectives. We analyzed the influence of the SD-AD data under four different influence factors, including the distance difference between the distance of the Argo and the satellite in $P_1$ and $P_2$ and angle difference, the distance between two Argo points, the sum of the distance of the Argo point and the eddy center in $P_1$ and $P_2$ and the difference in wind speed between two Argo points. The highlights of this study are listed below:

1.  The feasibility of validating the altimetry swath data by using the steric method. In this paper, we used in-situ observation data to analyze the feasibility of using a steric method to validate the interference altimetry sea level variability in different pixels. The result showed that when considering the distance difference between the distance of the Argo and the satellite in $P_1$ and $P_2$ and angle difference, the distance between two Argo points, the sum of the distance of the Argo point and the eddy center in $P_1$ and $P_2$ and the difference in wind speed between two Argo points, the relationship of the SD-AD has a highly corrected coefficient of 0.98, the RMSD was ~1.8 cm and the bias was ~0.6 cm. This proved that it is feasible to validate interferometric altimetry data using the steric method under these conditions.

2.  As we can see from Figures 4–10, when the distance difference between the distance of the Argo and the satellite in $P_1$ and $P_2$ were less than ~13 km, and in the same direction, the distance between two Argo points was less than ~120 km, the sum of the distance of the Argo point and the eddy center in $P_1$ and $P_2$ less than 220 km, difference in wind speed between two Argo points were less than ~1 m/s and the non-steric influence had a significant reduction. The relationship between the steric data and sea level data had a highly corrected coefficient of 0.98. This proved

that using the steric method to validate the sea level variability in different pixels is feasible, and the relationship needs to be studied in more detail in the future.

**Author Contributions:** Q.Z., F.Y. and G.C. designed the study, Q.Z. and F.Y. conducted the analysis, Q.Z. wrote the original draft, all authors contributed to final review and editing. All authors have read and agreed to the published version of the manuscript.

**Funding:** This research was funded by the following programs: (1) the Qingdao National Laboratory for Marine Science and Technology, grant number QNLM2016ORP0105; (2) the National key research and development program of China, grant number 2016YFC1402608, 2016YFC1401008; (3) the Marine S&T Fund of Shandong Province for Pilot National Laboratory for Marine Science and Technology (Qingdao), grant number 2018SDKJ0102-7.

**Acknowledgments:** We are thankful to the AVISO and IFREMER for providing the data.

**Conflicts of Interest:** The authors declare no conflict of interest.

## Abbreviations

The list of the abbreviation:

| | |
|---|---|
| ADT | Absolute Dynamic Topography |
| AE | Anticyclonic Eddy |
| AVISO | Archiving, Validation and Interpretation of Satellite Oceanographic |
| Cal/Val | Calibration and Validation |
| CMEMS | Copernicus Marine Environment Monitoring Service |
| ECMWF | European Centre for Medium Weather Forecasting |
| GDAC | Global Data Assembly Centers |
| IFREMER | French Research Institute for Exploitation of the Sea |
| MSSH | Mean Sea Surface Height |
| MSH | Mean Steric Height |
| NSH | Non-Steric Height |
| OW | Okubo–Weiss method |
| $P_1$ | Point 1 |
| $P_2$ | Point 2 |
| RMSD | Root Mean Square Deviation |
| SD-AD | Steric Data and Along-track Data |
| SH | Steric Height |
| SHA | Steric Height Anomaly |
| SLA | Sea Level Anomaly |
| SSH | Sea Surface Height |
| SWOT | Surface Water and Ocean Topography |
| T-S-P | Temperature-Salinity-Pressure |
| WOA 13 | World Ocean Atlas 2013 |
| 4D-Var | Four-Dimensional Variational Analysis |

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
