# Peer review of "The Difference of Sea Level Variability by Steric Height and Altimetry in the North Pacific"

_remotesensing, doi:10.3390/rs12030379_

Round 1
Reviewer 1 Report
Review of "The Difference of Sea Level Variability by Steric Height and Altimetry in the North Pacific" by Zhang et al.
Some parts of the manuscript have been improved, but I am not still satisfied with the manuscript resubmitted to Remote Sensing. Most of my comments in the previous review are not clarified. Therefore, I cannot recommend publication of this manuscript.
The reason why the authors evaluate the sea level difference between two separate points is still unclear. It is described in L41-43 that "the validation of interferometric altimeter is ... sea level variability between the different pixls in a swath". I cannot understand this context. Why cannot the single point evaluation be used even though interferometric altimeters such as SWOT also measures sea level at every measurement point? Do the interferometric altimeters observe sea level difference? Furthermore, specifically what is the benefit of the two points comparison in terms of evaluation of interferometric altimeter data?
The number of samples used in this study is very small. This is because the authors looked into sea level difference between two points and target area is limited to a small region to the south of the Kuroshio extension. This makes difficult to evaluate statistical significance of analysis results.
As I commented in the previous reviews, comparison between absolute value and anomaly such as SH-SLA and SHA-ADT do not make sense.
For example, if we take difference of sea level between positions (a) and (b), difference of ADT is written by
ADT(a) = SLA(a) + MDT(a), ADT(b) = SLA(b) + MDT(b)
--> ADT(a) - ADT(b) = SLA(a) - SLA(b) + MDT(a) - MDT(b) [eq.1]
where MDT is mean dynamic topography. Similarly, difference of steric height (SH) between the positions (a) and (b) is given by
SH(a) = SHA(a) + MSH(a), SH(b) = SHA(b) + MSH(b)
--> SH(a) - SH(b) = SHA(a) - SHA(b) + MSH(a) - MSH(b) [eq.2]
where SHA is steric height anomaly and MSH is mean steric height. As shown above, difference of SH between two separate points include
difference of mean field (MSH) as well as difference of anomalies (SHA). The difference of ADT also includes the difference of its mean field (MDT) between the two point, but SLA difference does not include difference in its mean field. That is why use of SLA data is not appropriate to compare with difference in SH. In this study, there is no large difference between for example SH-ADT and SH-SLA. This is because MSH(a)-MSH(b) in [eq.2] is not so large in the target region (south of the Kuroshio extension) although SH-SLA essentially does not make sense. If the comparison of SH-SLA is conducted near the Kuroshio Current, where there is a large gradient in mean steric height and thus MSH(a)-MSH(b) becomes large, the authors will find that there is large discrepancy in SH-SLA.
What do you mean by "distance different between the distance of Argo and the satellite in P1 and P2" in L251-252
Does it mean absolute value of L_argo - L_alt?
where L_argo denotes the distance between the two Argo points, and L_alt is the distance of two altimeter data.
Is it correct? Anyway it is difficult to understand this expression.
The authors suggest an importance of barotropic contributions from Fig.8, but there are other possible causes for the trend in Fig. 8b and Fig. 8d. One possible cause is the validity of the reference level of 900 m for the calculation of steric height. In general, velocity field in the Kuroshio extension exhibits deep structure and hence the assumption of no-motion at 900 m is not appropriate. Errors in mean fields for dynamic topography and steric height are also possible cause, and such errors would become clear when the distance of two points is large. Therefore, the conclusion that the linear trend is due to barotropic contribution seems optimistic and other possible causes should be discussed.
Why do the authors focus on wind speed in order to investigate barotropic contributions? What dynamics do the authors suppose? I commented the same question in the previous review, but I am not satisfied with your answer. Are barotropic sea level signals enhanced under strong wind speed condition such as tropical cyclones? In general, sea level changes as a result of convergence or divergence of surface velocity, which is caused by wind stress curl not wind speed.
How do the authors define the values of 13 km, 120 degree, 120 km, and 220 km in L355-357. They seem too subjective.
Reviewer 2 Report
The author's have adequately addressed my previous review comments.
Author Response
Thanks for your constructive and helpful comments and for your approval of our modification.
Reviewer 3 Report
Review Report for RESUBMISSION of the manuscript The Difference of Sea Level Variability by Steric Height and Altimetry
in the North Pacific by Qianran Zhang, Fangjie Yu and Ge Chen
I would like to thank the authors for providing detailed answers to my comments and modifying the manuscript accordingly. I confirm that the manuscript has improved a lot, though english has to be double checked again. Sorry if I may look too meticulous, but I wonder if the authors have considered the help of a native English speaker to revise the whole text. I think it could make this interesting paper more valuable. Moreover, I would suggest to explicitly indicate the line numbers where the corrections can be found in the revised manuscript, this would facilitate the Reviewers’ work. In the end, I would like to thank the authors for clarifying my questions on the wind speed, it was probably my mistake to misunderstand the text.
Some minor comments are attached.
Line 145: “this paper, we using” should be changed to “ in this paper we used”.
Line 170: “ as a set of data which” should be changed to “ as a set of data for which…”
Line 175:” the eddy which detect “ should be changed to “ the eddy which was detected”
Line 178: “There were four height variables” is unclear, please rephrase
Lines 180-183: Caption of figure 4, please rephrase as “The maps a-c respectively indicate data groups N. 6, 13 and 24, that will be discussed in the next section”
Line 184: “In the next section, we discuss two different approaches for analysing the impact of different factors on the SD-AD data to eliminate the impact” is unclear to me. Please rephrase.
Lines 190-191: “can be considered as the same point better” should be changed to “can be approximately considered as lying on the same point”
Lines 252-255: “Regardless of the SH or SHA, the relationship of the SD-AD had the similar trend that the results after linear regression were different in terms of slope, no matter for SH and SHA the ADT data performed better than the SLA data when the distance difference was smaller”. I think this sentence still needs to be improved. To me, the meaning is not clear.
Lines 285-286: “…became shorter, the results of the SD-AD under the satellite data selecting the ADT data or the SLA data are more similar which proved that the geoid and MSSH were similar in a small range” should be changed to : “…became shorter, the results of the SD-AD under the satellite data selecting the ADT data or the SLA data are more similar. This proved that the geoid and MSSH were similar in a small range”
Line 323: “when the distance between the Argo points and the eddy centre smaller than 220 km” should be changed to “when the distance between the Argo points and the eddy centre is smaller than 220 km”
Round 2
Reviewer 1 Report
None
Author Response
Thanks for your constructive and helpful comments.
This manuscript is a resubmission of an earlier submission. The following is a list of the peer review reports and author responses from that submission.
Round 1
Reviewer 1 Report
Second review of "The Difference of Sea Level Variability by Steric Height and Altimetry in the North Pacific" by Zhang et al.
The authors' revisions have clarified some of my questions in the previous review comments. However, I am still not satisfied with the manuscript.
The largest difference from previous study is that this study validates sea level difference at two separated points in contrast to single point validation in previous studies. The adopted method, however, is essentially the same as the single point validation.
As I commented in the previous review, comparison between absolute value and anomaly such as SH-SLA (e.g., Fig. 5cd), SHA-ADT (e.g., Fig. 6ab) do not make sense. This is because difference of SH or ADT at two separated points includes difference of mean field, while difference of SLA or SHA at two points does not include it. Comparisons should be done between SH-ADT or SHA-SLA.
Section 3.2.3 also does not make sense. High-frequency signals in altimeter data are filtered out. If the authors consider barotropic signals, the authors should look into seasonal variations, which have relatively large barotropic contributions especially at high latitudes.
The number of sampling data is very small. Therefore, the authors should evaluate statistical significance.
Discussions on physical meaning of validated results are very poor throughout the manuscript.
There seems to be errors in figure legends. The authors should carefully check them.
SH-ADT in Fig. 5d should be AH-SLA.
SH and SH-SLA in Fig. 6b should be SHA and SHA-SLA.
There are still English errors.
Reviewer 2 Report
Review of the manuscript “The difference of sea level variability by steric height and altimetry in the North Pacific”
In this manuscript, relationship between Argo profiling float-based steric data (SD) and satellite altimetry-based along-track data (AD) around a typical mesoscale eddy is analyzed to assess feasibility of the steric method of (particularly future) altimetry data Cal/Val. The difference (SD-AD) is analyzed as a function of distance between two Argo positions, between Argo positions and the eddy center, and wind, supporting feasibility to validate future interferometric altimetry data using the steric method. The topic is of significance considering the future altimetry data Cal/Val, and this manuscript provides an interesting take on a well-known relationship between SD and AD. However, there are some major as well as minor questions and comments. Thus, my recommendation is major revision.
Overall issue: reasoning for the difference between SD and AD was not appropriately delivered. What explain(s) the difference, barotropic contribution of sea surface height change or bottom pressure or errors in satellite altimetry measurements or errors in satellite altimeter range corrections, or errors in Argo profiling float measurements? All these sources producing the difference (SD-AD) have pertinent characteristics and scale of variability, and should be incorporated into the Cal/Val analysis. Line 22 and others: corrected coefficient? Do you mean correlation coefficient? Line 46: In “wang [12]”, the first author’s last name of cited reference should be capitalized. 5, 6, and 7: the difference (SD-AD) occasionally exceeds 10 cm. Why are the two sea surface heights so different? Main reason(s) for these extreme differences should be discussed. Oceanic mesoscale eddies may have different baroclinicity having different amplitude or sea surface height difference between center and edge. It would not be a simple linear function of distance from the center but combination of distance from the center, eddy amplitude, eddy size, etc. Lines 258-260: Is the relationship uncontrollable? Is these increased difference unreasonable? Captions of Fig. 8b and 8d: not SD-AD but the absolute. Line 262: Correction? Do you mean correlation?
Reviewer 3 Report
The authors compare sea height data dervied from different sources (altimetric and ARGO/steric data) and with different reference level (geoid, mean SSH etc.). They propose that steric data could be used to validate altimetric data, and they diagnose some conditions where this comparison is most valid, based on a combination of the separation distance of the two observations, the wind speed, and the proximity of the observations to the eddy centre (the eddies are largely responsible for the sea height variations at this scale).
This paper contains results that could be useful for a wider audience, and timely in the context of upcoming new data sets (e.g. SWOT). However, I think the manuscript needs to be substantially improved, particularly changes should be made to (1) the use and explanation of acronyms and abbreviations, (2) the presentation of results in figures [many contain errors/ambiguities], and (3) the discussion. My comments are as follows:
Abstract - It is nice to use SD-AD here as it is general and easy to understand, but skipping ahead to figures and discussion this notation is not used consistently (i.e. SHA-ADT, SH-ADT, SH-SLA, SLA-ADT^{not SD-AD} and SHA-SLA). Could you find a way to present SD-AD throughout the manuscript with some distinction between different comparisons of the two. For example, SD-AD (SHA-ADT) or (SD-AD)_{SHA-ADT}.
Line 30 - 'In' should be 'Since'
Figure 1 - It would be helpful to have a reference box on (a) to show the size and location of domain (b)
Line 83 - Explain in text the time period of choice.
Line 94 - Sentence beginning 'Figure 3...' is unclear.
Line 97 - Was the satellite data chosen to be exactly the same time period? What are the temporal offsets?
Figure 2 - Since you are providing a schematic you should clearly include the acronyms used in this paper. SH, SHA, MSH, NSH etc are not clearly defined in this figure. Altimetric range and satellite altitude are also not discussed sufficiently, and the meaning of geoid, ellipsoid, and their differences should be clarified.
Line 106 - Meaning of sentence beginning 'To ensure the conditions...' is unclear
Line 116 - This statement is not clear from Figure 2
Section 2.3 - It is confusing to use subscript 1/2 for atmosphere/bottom pressure and subscript A/B for distinction of positions. Switching those round would be more intuitive for readers.
Equation 11 - Could you put SH_A-SH_B on the far right of the equation so it can be compared to the same term in the other ARGO variable D_{SHA} (next equation)
Line 135 - What is NSH?
Line 136 - What geoid is used as a reference?
Figure 5 - You compare all the subpanels in text but these comparisons would be aided by condensing/adjusting the figures. I would suggest putting AH, ADT and SLA on the same figure, they are already different cooloured symbols, and this would essentially mean condensing (c) with (a) and condensing (d) with (b).
Figure 5 - You should also change the colour of the SLA-ADT line to match the SLA symbol colours (repeat for SH-ADT line color).
Figure 5 - You should also add a zero reference line
Figure 5 - Your captions in (d) for the line appears to be wrong.
Line 168 - What does 'at latitude and longitude' mean?
Line 183 - 'Better' than what? Is this for SLA and/or ADT?
Figure 6 - Is top-left line SH-ADT or SHA-ADT, legend is confusing.
Figure 6 - Is bottom right figure plotting SH or SHA (legend is confusing).
Figure 6 - Add reference axes at y=0
Figure 6 - Could you add faded lines repeating lines from Figure 5 (for SH) to aid comparison of SH comparison errors relative SHA comparison errors
Lines 221-223 - Please clarify this sentence.
Line 221 - What about the data at ~8km, which appears to be even more off-the-linear-fit than data No. 13.
Figure 7 - Using AD (y-axis label) and ADT (captions) in the same figure is confusing.
Figure 8 - I believe the dots are not SH-AD (they are AD-AD). So the y-axis labels and caption are confusing. What is the purpose of dots for discussion?
Line 282 - Explain this statement (preferably in terms of processes/dynamics)
Line 283 - Was this ECMWF data from exactly the same time period? Is it temporally/spatially averaged, as I suspect the small-scale ECMWF variability could be large and decorrelated from the true wind speed variability.
Line 360 - The role of direction is not clear from Figures 7-10 alone.
Reviewer 4 Report
Review of the manuscript: “The Difference of Sea Level Variability by Steric Height and Altimetry in the North Pacific”
By Zhang et al.
I acknowledge the authors for this work, I think it is interesting and also necessary to point out how to validate wide-swath altimetry data. However, the text requires major revisions, sometimes it is very hard to get the concepts and methodologies used. If possible, I would recommend a revision from a native English speaker. Also, some of the methodologies of investigation are not clearly defined. In the end, I recommend this paper for publication after an extensive major revision. My general comment is detailed in the following list of suggestions/corrections:
Abstract
Line 14: please change “invented” with “developed”
Introduction
Line 44-45: “ Now, existing Cal/Val methods, which include the offshore method [11], the steric method and more 44 indirect method, do not have an exact method”. This sentence is confusing. It should be completely rephrased. More simply, I would point-out that there are different Cal/Val approaches and I would list differences between these approaches and how they compare with the one you propose in your study;Line 45: “the steric method was” should be changed to “the steric method consisted in”;
Line 45: What do the authors mean when they write:” the steric method and more indirect method, do not have an exact method”?. It is not clear, please rephrase.
Line 46: please change format of the citation (e.g. Wang et al. 2018). The same comment is valid for all other citations in the manuscript. They should all appear in the format “First_author et al., year”;
Line 62: please change feasible with feasibility;
Line 67: please rephrase: “We relied on 17 years of Argo data that…” instead of “there was”;
Materials and Methods
Line 74: Please give a more detailed description of the Argo program, what is the aim of the initiative? Which is the typical vertical sampling range? Just a couple of additional information, I think this is helpful;Line 77: IFREMER’s should be changed to IFREMER. Moreover, what is GDAC? Please define all acronyms, or even better, create a section at the end of the paper with the list of all acronyms in the paper. I think It can help the reader;
Line 84: Unless I am mistaken, you did not mention how you derived the eddy track and Eddy area. Is it coming from a catalogue or maybe the authors relied on an eddy tracking software? What do you mean for eddy area? Is it based on the knowledge of the eddy dynamical radius or derived from Okubo-Weiss parameters? Which are the mean Eddy Characteristics? Radius, Swirl velocity? Are the Eddy characteristics compatible with the use of gridded altimetry products? I think this should be stated in the text;
Line 89: CMEMS. See my comment for Line 77;
Lines 89-94: This paragraph is a bit confusing to me. Please try to make it easier to read simply recalling that a satellite measures SSH above the reference ellipsoid, but the interesting variable for oceanographic applications is the Dynamic Topography (sea level above geoid) which is the expression of the transient part of the surface geostrophic motion;
Line 125: I have a question on the use of p2 = 900 m. Is this meaning that the 900 m depth is representative of your ocean bottom pressure level? Can you please further clarify this choice?
Line 130: SHA is not defined. Please see my comment for Line 77;
Line 135: NSH. Is this a mistake? Maybe this is MSH?
Line 135-143: honestly I struggled a little bit to get the meaning of these lines.
-First of all, it seems you have been using gridded altimetry products. This is not mentioned in the data description, please add a sentence on that.
-what do you mean exactely with “ the trends of the MSSH and geoid coincide well”? Where did you show this result?
- I cannot entirely get the meaning of the groups of data. Are you saying that you have been looking for Along-track SLA and Argo profiles at the eddy periphery (or in proximity of the eddy radius)? If you had to give an example of point A and point B (the one indicated in equations 9-12, where would they lie in your figure 4? The “groups of data” indicate the number of available satellite and Argo observations along the eddy positions? Lines 153-154: I think you are just saying that when Argo and Satellite data are nearby in space and time, the representativity error is reduced. If so, rephrase these lines to clarify the text;
Lines 169-185. These lines are also a bit confusing. I think I would organize the numerical results in a table, as in the example below (please also double check the significant decimal digits, sometimes they are different):
|
Distances Argo//Along-track Sat |
RMS |
BIAS |
Correlation coefficient |
|||
|
0.2 degrees |
ADT 5.25 |
SLA 4.42 |
ADT 1.54 |
SLA 1.18 |
ADT 0.8858 |
SLA 0.9034 |
|
0.15 degrees/4 hours |
ADT 4.07 |
SLA 3.87 |
ADT 0.5379 |
SLA 0.5687 |
ADT 0.9241 |
SLA 0.9292 |
In the end, at line 176 I would write “The RMSD and Bias are reduced” instead of greatly improved.
Lines 186-194: here, I would also organize the numerical results (RMS and BIAS) in a table. Also, please report even the numerical results for the “more restrictive” spatial-temporal distance requirements;
Line 207: please remove greatly;
Line 212-216: In my opinion “To obtain better SD-AD results, we used more factors to analyse the results. Figure 7 shows the relationship between the SD-AD under the condition of distances difference of the distance between each Argo position and altimetry position. Regardless of the SH or SHA, the relationship with the SH-AD had a similar trend, and the SHA datasets had a clearer trend because the slope of the SHA was larger than that of the SH” should be rephrased as follows: “To obtain better SD-AD results, we used more factors to analyze the results. Figure 7 shows the relationship between the SD-AD as a function of the ARGO and Satellite data relative distances. Regardless of the SH or SHA, the relationship with the SH-AD had a similar trend. The SHA datasets had a clearer trend, being the SHA slope larger than the SH one.
Line 219: “regarding the results of the SHA-AD” should be changed to “for the SHA-AD case”;
Line 221-223: could you please add a graphical explanation for the group points 13? The description on the direction of the Argo points relative to the altimeter data is unclear? Would it be possible to rephrase these sentences? I think this is crucial for the paper; Lines 239-247: I think that this paragraph also needs extensive rephrasing. I could only retain that for increasing relative distances between Argo points the SH-ADT and SH-SLA tend to increase linearly. Then, what do you mean when you say that “When the amount of SD data was larger than that of altimetry data, the results were more concentrated”? Also, what do you mean with “When a linear regression analysis was performed on the distance among Argo points, regarding the result of the SD-AD, the ADT data performed well when the condition was less restrictive”? Which condition are you referring to? Can you please clarify all this?
Line 256-260: What do you mean with the statement “The result of the SD-AD is larger than 2 cm, which is not sufficient”? Please clarify;
Line 275-277: please further clarify this sentence, it is not clear;
Line 287: please modify as “…clearer results than the SD-AD using SH”;
Lin 302-309: In general, for the numerical results illustrated here, I would add another table, see my comment for Lines 169-185;
Line 308: I would modify as:” these values represent a further improvement with respect to previous analyses”
Line 319-322: please further clarify the sentences at these lines. To my understanding it should be something like: “Meanwhile, considering that the ocean variability has the same trend in a small area, the direction between the Argo and altimeter measurements sites becomes another important factor”. However, I would like to see a graphical representation of this, as already pointed out at lines 221-223;
Line 329: “with that of our previous research”. Are you referring to another paper or to the previous analyses in this paper? This has to be clearly stated;
Line 331-332: Please rephrase these two lines, they are not clear;
Line 349: I would write as follows:” The highlights of this study are listed below:”;
Line 362-363: I have one major comment related to this statement. You are indicating that the the optimal conditions for validating wide-swath altimetry data via steric measurements is to have a wind speed ~1m/s. In a recent paper appeared in Remote Sensing àCiani, D., Santoleri, R., Liberti, G. L., Prigent, C., Donlon, C., & Buongiorno Nardelli, B. (2019). Copernicus Imaging Microwave Radiometer (CIMR) Benefits for the Copernicus Level 4 Sea-Surface Salinity Processing Chain. Remote Sensing, 11(15), 1818. The authors analyzed the behavior of the Mean wind speed at the sea-surface using AMSR-2 satellite data (see their figure 10). They showed that, on average, during the year 2016 the mean wind speed was always larger than 5 m/s. Can you please justify how your 1 m/s criterion can be compatible with such a scenario?
Figures
Figure 1:please change the legend to the colorbar of figure 1b as follows: SLA (m) and not SLA/m. Still in figure 1b, please write longitude (°E), latitude (°N) and not longitude/°E, latitude/°N;Figure 4. Please insert the legend aside the colorbar: SLA (m); What do the three maps stand for? Are they representing three different dates? Please clarify this;
Figure 5: I do not understand the legends of the panels in the figure. First of all I would write “difference” instead of difference. But I don’t think that the y axis represents the differences between the in-situ and the Argo data (only 5 to 20 cm apart). Instead, I think these are the differences in height when you consider Argo or satellite Along-track data. So, probably the legend should be modified, do you agree with that?
Figure 6: same comment as in figure 5. Moreover, is the legend of panel (d) wrong? Maybe you meant SHA, SLA, SHA-SLA?
Figure 7: same comment as in figure 5. Moreover, could you please add a graphical explanation for the group points 13? The description on the direction of the Argo points relative to the altimeter data is unclear. In the caption, I would change “under the condition” with “as a function of”;
Figure 8: same comment as in figure 5. Moreover, In the caption I would change “under the condition” with “as a function of”; Figure 9: same comment as in figure 5. Moreover, In the caption I would change “under the condition” with “as a function of”;
Figure 10: same comment as in figure 5. Moreover, In the caption I would change “under the condition” with “as a function of”;